# SITTO: Single-Image Textured Mesh Reconstruction through Test-Time Optimization

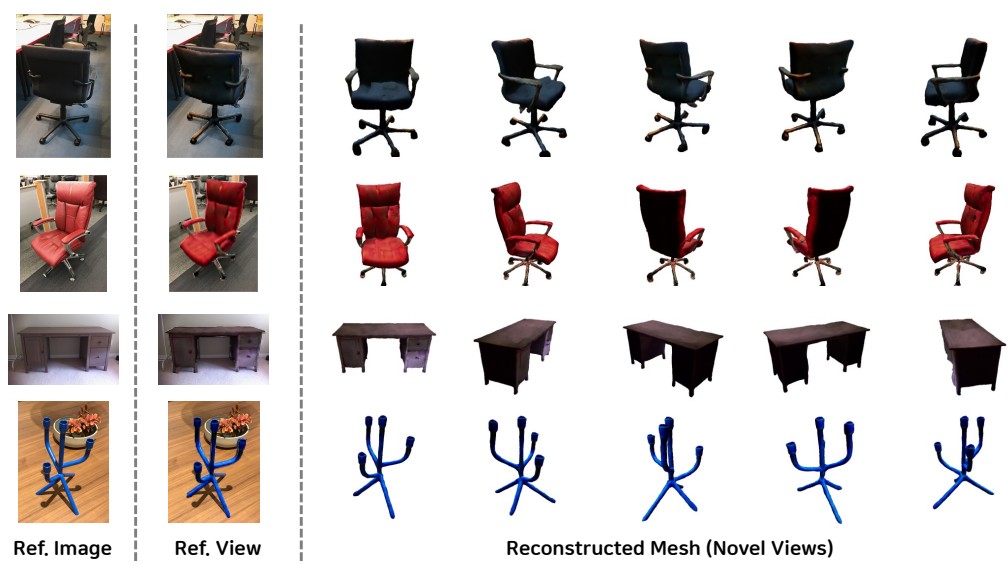

Figure 1: **Qualitative sample results of** SITTO. Our method reconstructs high-fidelity geometry and realistic textures from a single image. [Ref. Image] The first column shows the reference sample image provided as a single input. [Ref. View] The second column shows the overlayed rendering result of the reconstruction obtained from SITTO at the reference viewpoint. [Novel Views] The remaining columns show the results of our method for novel viewpoints.

## ABSTRACT

Reconstruction of a 3D textured mesh from a single image has been a long-standing and challenging problem. To address this challenge, we aim to leverage existing feed-forward-based models designed for predicting shape (*i.e.*, textureless mesh) from a single image. However, there are difficulties that have to be overcome. Firstly, methods that estimate shape using feed-forward approaches cannot always guarantee high-quality results. A test-time optimization technique with feedback loops specified to each target object instance is necessary to apply these methods practically. To tackle this, we unlock the recent advancements in multi-view diffusion models, showing impressive multi-view image generation performances. Nonetheless, there are challenges associated with utilizing diffusion models. Specifically, it is crucial to estimate the viewpoint of the given reference image (*i.e.*, its elevation and azimuth angles) and sample relative viewpoints from the reference viewpoint. We solely employ neural mesh representation and texture optimization to optimize training efficiency in terms of time and memory complexity. SITTO tackles these challenges by introducing an automatic pipeline for monocular 3D textured mesh reconstruction with test-time optimization. Our method demonstrates impressive results in fine-grained geometry details and the generation of realistic texture appearances.

## 1 INTRODUCTION

There has been a growing effort among researchers and developers to create virtual reality systems designed for virtual communication or entertainment purposes. The generation of 3D content is considered a pivotal technique to realize extended reality, including applications for Augmented Reality (AR) and Virtual Reality (VR) systems. Nevertheless, generation methods of traditional 3D contents and assets require expert knowledge and skill within the field, accompanied by substantial labor and time complexity. Furthermore, they are not scalable to extend to large-scale contents.

To address these challenges, there have been trials in the realm of generating 3D content from single images, especially for 3D objects. Most of the existing studies focused on the generation of 3D meshes solely through feed-forward methods when provided with only a single image (Gkioxari et al., 2019; Pan et al., 2019; Nie et al., 2020). Despite using 3D shape data for training, obtaining 3D meshes with high-fidelity geometry poses a complex challenge. Furthermore, these efforts only yield geometric information with no texture details. Recently, the field of 3D content generation, particularly based on per-instance through optimization methods, has shown remarkable advancements (Mildenhall et al., 2021; Fridovich-Keil et al., 2023; Chen et al., 2022; Müller et al., 2022). These advancements have been driven by the development of methods that rely on Neural Radiance Fields (NeRFs) (Mildenhall et al., 2021) serving as foundational 3D representations. NeRFs employ multiple viewpoints to reconstruct 3D representations through per-instance optimization.

Additionally, recent research has emerged to generate 3D structures from text or images (Poole et al., 2023; Lin et al., 2023; Chen et al., 2023). Generating 3D models from just a single image is a well-known and challenging problem due to the need to maintain consistency between geometric details within unseen views. However, the recent success of 2D generative models (Rombach et al., 2022; Saharia et al., 2022; Balaji et al., 2022; Ramesh et al., 2021; 2022) has made it possible to use learned prior knowledge for this task. Just as humans can imagine what the unseen sides of objects look like based on their experiences, 2D generative models apply their extensively trained data to play a similar role in the generation process. Many arts have worked on unlocking the capabilities of these models to conduct 3D lifting using 2D generative models (Poole et al., 2023; Lin et al., 2023; Chen et al., 2023; Melas-Kyriazi et al., 2023; Deng et al., 2023; Tang et al., 2023; Liu et al., 2023a; Qian et al., 2023). This process is crucial for achieving high-fidelity 3D generation results.

However, regarding using NeRFs in real-world industries, they face practical challenges. This is mainly because they use a type of representation that does not easily convert into the mesh representations used in industry. Although one can use a post-hoc method to convert them into meshes using techniques like Marching Cubes (Lorensen & Cline, 1998), doing so produces degraded mesh quality which does not reflect the quality of the NeRF result. Moreover, NeRF-based methods demand a substantial amount of memory and time footprint to train detailed 3D models, which, as explained in **?**, imposes constraints on the resulting resolution. Various approaches (Lin et al., 2023; Chen et al., 2023; Metzer et al., 2023) have been proposed to overcome these challenges by directly representing 3D objects in more practical mesh representations, where Lin et al. (2023) rely on coarse NeRF reconstruction as an initialization and Metzer et al. (2023); Chen et al. (2023) a user-provided template mesh as the starting point for subsequent mesh optimization. In essence, these methods strongly rely on having a good initial shape for optimization; otherwise, the results obtained from the approaches are significantly degraded with arbitrary initialization.

Therefore, we propose an instance-wise and perceptually well-aligned 3D shape and texture reconstruction method, where our method generates holistic object shape and texture of each instance in the scene given a single image. The key idea is to integrate feed-forward mesh generation that can quickly create 3D initial shapes, followed by optimization-based feedback loops in test time. In this way, we can start our test-time optimization with a nice initial shape and produce high-quality meshes automatically for each instance. We can acquire not only geometry but also texture information that incorporates physically based rendering. This comprehensive approach enables textured 3D mesh reconstruction from a single image, all within a single automatic pipeline. Our key contributions are summarized as follows:

- Introduce an automated pipeline `SITTO` from initial shape generation to create textured mesh reconstruction with test-time optimization.

- Explore how to unlock the capacity of multi-view aware diffusion capacities with only mesh-based approaches without using implicit representations.
- Show high-fidelity textured mesh generation results, particularly in scenarios involving complex, non-circular structures.

## 2 RELATED WORK

The pursuit of reconstructing 3D contents from single images has been a longstanding endeavor. In this section, we briefly review the closely related work with our work.

**Feed-forward neural network based 3D mesh reconstruction.** Gkioxari et al. (2019) extends the conventional 2D perception method known as Mask R-CNN (He et al., 2017) by introducing a mesh prediction head, called Mesh-RCNN. This research aimed to reconstruct object shapes in the regions of interest (RoIs) aligned with the input image, effectively bridging the gap between advanced 2D perception and 3D understanding, and facilitating the development of a unified architecture adaptable to both synthetic and real-world scenarios. Later, Nie et al. (2020), in a similar vein, extend the framework to bridge scene understanding and object reconstruction by integrating the tasks of room layouts, 3D object bounding boxes, and textureless object mesh reconstruction from single images concurrently.

Both of these works address the limitations of a solely 2D-focused or a primarily synthetic-data-oriented approach, providing a powerful foundation for holistic scene understanding. However, they are subject to certain constraints, such as the inability to guarantee high-quality shape and the lack of consideration for texture information. These limitations mainly come from the challenge of single-view 3D reconstruction for feed-forward frameworks, as instance detection, classification, and segmentation have to be addressed along with reconstruction. Our observation is that, although these methods are susceptible to the performance of each sub-task and the quality of state-of-the-art results found lacking, they seem to have reasonable perceptual capability to produce plausible initial shapes from a single image. This motivates us to develop a test-time optimization module that carries out per-instance optimization to exploit additional information from the given test image. Also, while Gkioxari et al. (2019); Nie et al. (2020) focus on textureless mesh reconstruction, our work reconstructs texture meshes of objects that successfully comply with given images. A concurrent work (Liu et al., 2023a) and Jun & Nichol (2023) propose feed-forward models requiring training on a textured 3D training dataset and focus on generation rather than reconstruction.

**Optimization based 3D generation.** Recently, there has been a noticeable trend in 3D shape generation research, with a growing focus on optimization methods for fitting shapes on a per-instance basis, *e.g.*, recent text-to-3D methods (Poole et al., 2023; Lin et al., 2023; Metzer et al., 2023; Chen et al., 2023; Wang et al., 2023), rather than relying solely on feed-forward approaches. This trend is influenced by the advancements in 2D generative diffusion models, such as text-to-image (Rombach et al., 2022; Balaji et al., 2022; Saharia et al., 2022; Ramesh et al., 2021; 2022) and image-to-image models (Liu et al., 2023b), as strong prior knowledge of 3D shape understanding. A representative advancement is made by the use of the pre-trained 2D diffusion model as a learned generative prior for rendered views, called Score Distillation Sampling (SDS) loss (Poole et al., 2023). The SDS enables generating 3D objects without a separate 3D training dataset but by optimizing a common 3D representation, *e.g.*, NeRF (Mildenhall et al., 2021), such that its rendered views comply with the 2D pre-trained diffusion model through SDS.

Motivated by the success of SDS (Poole et al., 2023), there have been recent efforts for single view 3D generation methods (Tang et al., 2023; Qian et al., 2023; Melas-Kyriazi et al., 2023). Their focus is mainly on the perceptually plausible generation task rather than the accurate reconstruction (Gkioxari et al., 2019; Nie et al., 2020). Melas-Kyriazi et al. (2023) combine the idea of the text-to-3D (Poole et al., 2023) with textual inversion (Gal et al., 2022) to represent the reference image as a text token. Qian et al. (2023) extend Melas-Kyriazi et al. (2023) by additionally incorporating a separately pre-trained 3D diffusion model as prior, which requires separate training the prior with large-scale 3D objects. Tang et al. (2023) employ coarse-to-fine optimization strategies with NeRF representation. High memory complexity of NeRF representation poses challenges when aiming for high-resolution 3D shapes (Lin et al., 2023), which demands coarse-to-fine-like multi-stage approaches in Tang et al. (2023). Also, NeRF representation itself appears plausible, but in practice, it is not readily converted into meshes directly, leading to degraded mesh quality in the final product.

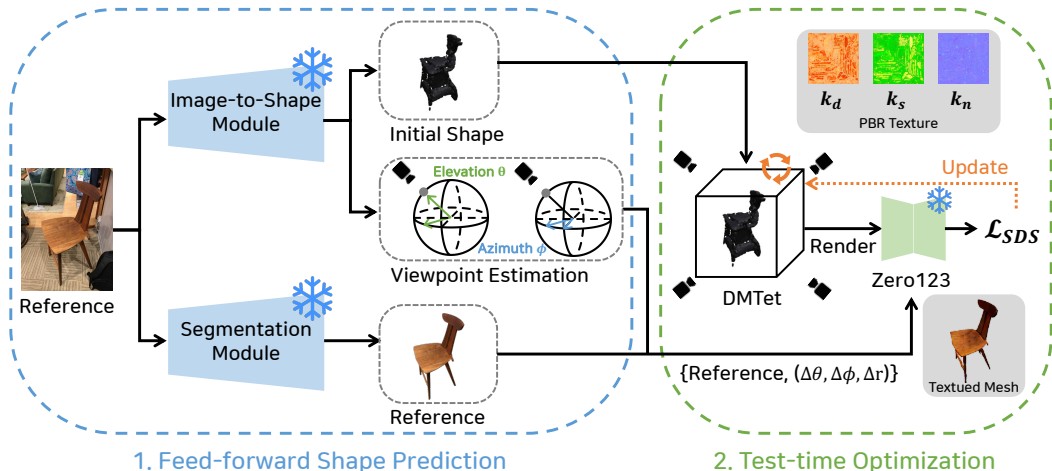

Figure 2: **Overview of** SITTO. We propose a two-stage automated pipeline to reconstruct a textured 3D mesh from an image. We obtain a coarse shape of an object through a feed-forward shape prediction stage. Subsequently, we conduct a test-time optimization, where we utilize a multi-view aware image diffusion model to reconstruct a texture and the fine details of the object meshes.

To produce a desirable mesh representation as a final output, recent work, *e.g.*, (Lin et al., 2023; Chen et al., 2023) in text-to-3D and (Qian et al., 2023) in image-to-3D, has started to adopt signed distance function (SDF) based mesh representations like DMTet (Shen et al., 2021). However, in our preliminary study, we observed that mesh representations are prone to the initialization of the shape. To sidestep this challenge, Lin et al. (2023); Qian et al. (2023) exploit coarsely trained NeRF to obtain coarse mesh or user-provided mesh templates (Chen et al., 2023). We directly optimize SDF through DMTet initialized from the feed-forward shape prediction (Gkioxari et al., 2019; Nie et al., 2020), which leads to stable and finely detailed textured meshes.

## 3 METHOD

We first provide an explanation of the flow of the feed-forward shape generation pipeline, also referred to as *Image-to-Shape* in Sec. 3.1. Following that, we describe how to adopt multi-view aware image diffusion model in the test-time optimization for reconstructing 3D textured meshes in Sec. 3.2, which also call *Shape-to-3D*. Finally, we explain how to conduct optimization with only mesh representation along with textured surface in Sec. 3.3.

### 3.1 IMAGE-TO-SHAPE: FEED-FORWARD SHAPE GENERATION

In this stage, we infer the initial shape and camera viewpoint from single input image. The purpose of curating these is to facilitate the optimization of the 3D textured mesh. Because to leverage the multi-view aware capacity of Zero123 (Liu et al., 2023b), we have to know the camera position of the reference image, which is also called reference angles. For this, we utilize Total3DUnderstanding (Nie et al., 2020) as our *Image-to-Shape* architecture because it is suitable for the naive initial shape and viewpoint estimation (*e.g.*, elevation and azimuth angles). Note that the choice of *Image-to-Shape* method is not bound to the specific model. We skip the mesh refinement stage, which includes the edge classifier and boundary refinement stages. During the mesh refinement stage, some edges are cut, and the resulting mesh with cut edges can be unstable when initializing Signed Distance Function (SDF) representations, especially when the mesh surfaces tend to be not watertight. During the test-time optimization stage, we undertake the optimization of estimated values for elevation and azimuth angles. This is beneficial because the estimated values are not exact and require further refinement for precise results.

In this stage, we have to curate our input image to be segmented and filled with a white background because Zero123 only works with this image form. We leverage (Eftekhar et al., 2021) or (Kirillov

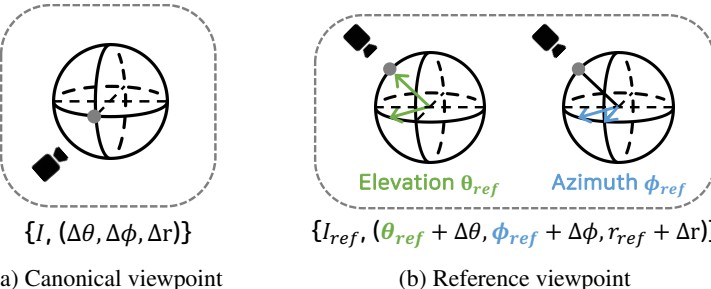

{$I, (\Delta\theta, \Delta\phi, \Delta r)$}

(a) Canonical viewpoint

{$I_{ref}, (\theta_{ref} + \Delta\theta, \phi_{ref} + \Delta\phi, r_{ref} + \Delta r)$}

(b) Reference viewpoint

Figure 3: **Viewpoint estimation.** (a) Zero123 assumes the canonical spherical coordinates, meaning the object is centered at the coordinate system's origin, and the camera consistently points toward this origin. In these canonical spherical coordinates, the initial camera viewpoint is positioned at the canonical viewpoint, where both the elevation and azimuth angles are set to $0°$. (b) To leverage the Zero123's of multi-view generation capability, we need to know the reference viewpoint, involving the elevation and azimuth angles, from the reference image. This necessitates the calculation of the relative angles between the reference viewpoint and the newly sampled viewpoint.

et al., 2023) as the segmentation modules. Because our 3D mesh reconstruction quality has to be bounded by the image segmentation quality, the best performance segmentation model have to be selected. Because we will optimize the reference image to canonical 3-dimensional space, we conduct recentering and resizing segmented images.

## 3.2 SHAPE-TO-3D: TEST-TIME OPTIMIZATION FOR 3D TEXTURED MESH RECONSTRUCTION USING MULTI-VIEW AWARE DIFFUSION MODEL

Utilizing the initial shape generated through the *Image-to-Shape* process, we proceed to initialize our 3-dimensional representation spaces, specifically employing the DMTet (Shen et al., 2021). From this shape initially, we optimize to *3D*, which is our final target and means textured mesh outputs, which we call *Shape-to-3D* process.

**Multi-view aware image diffusion model.** To get our target textured mesh, we adapt test-time optimization with multi-view diffusion model (Liu et al., 2023b). There are many attempts to unlock the pre-trained 2D diffusion models at other tasks, such as 3D or video generations. We also need to try to leverage the ability of 2D diffusion models, especially those trained with multi-view information. One of the most notable methods is Zero123 trained with Objaverse (Deitke et al., 2023b;a), which is a large-scale object dataset. Zero123 operates under the assumption of a canonical spherical coordinate system, where the camera is positioned on the surface of the sphere and directed towards the origin of the coordinates. We represent $\theta, \phi$, and $r$ for elevation angle, azimuth angle, and radius. If one image is given with viewpoint $(\theta_{ref}, \phi_{ref}, r_{ref})$, and we want to generate another view with viewpoint $(\theta_{new}, \phi_{new}, r_{new})$. Given that Zero123 is a view-conditioned diffusion model, it necessitates the provision of $(\theta_{new} - \theta_{ref}, \phi_{new} - \phi_{ref}, r_{new} - r_{ref})$ as a condition to the diffusion models, which is equivalent to relative viewpoint $(\Delta\theta, \Delta\phi, \Delta r)$ Fig. 3b. With an estimated viewpoint of the reference image from *Image-to-Shape* stage, we can harness the view generation capacity for 3D reconstruction. Zero123 demonstrates an impressive capability to generate multi-view images from a given viewpoint. However, it has the ability to preserve content consistency rather than accurate geometric consistency. To exploit its multi-view capabilities without precise geometric consistency, we opt for using SDS (score distillation sampling) loss (Poole et al., 2023) instead of color loss (Mildenhall et al., 2021). We employ color loss only at the given reference view image.

**Test-time optimization for 3D reconstruction.** We employ DMTet as our 3D representation, which is characterized by two essential features. It encompasses a deformable tetrahedral grid used to represent 3D shapes and a differentiable marching tetrahedral (MT) layer designed to extract explicit triangular meshes. This representation has vertices $V_T$ in the tetrahedral grid $T$, and can be expressed a $(V_T, T)$. At each vertex $v_i \in V_T$, it predicts the signed distance function (SDF) $s(v_i)$ and vertex deformation offset $\Delta v_i$ with hash-grid positional encoding (Müller et al., 2022) function $\tau$ as follows:

$$[s(v_i), \Delta v_i] = \Theta(\tau(v_i); \theta) \tag{1}$$

where MLP network $\Theta$ has the parameters $\theta$. Before starting to optimize the target object from the reference image, we first initialize DMTet with the initial shape obtained from *Image-to-Shape*. From this initial shape, we sample a set of points $p_i \in \mathbb{R}^3$ where $p_i$ represents a point in $P$ that is the closest to the shape surfaces. We then compute their SDF values $SDF(p_i) \in \mathbb{R}$ as follows:

$$\mathcal{L}_{SDF} = \sum_{p_i \in P} \|s(\tau(p_i); \theta) - SDF(p_i)\|_2^2. \tag{2}$$

Using the pre-optimized network $\Theta$ and a differentiable render $R$ (*e.g.*, (Laine et al., 2020)), we can generate the RGB rendering image $\boldsymbol{x}$ as $\boldsymbol{x} = R(\theta, c)$, where $c$ represents the sampled camera viewpoint. We randomly sample camera viewpoints within the range of [-45°, 45°] for the elevation and [0°, 360°] for the azimuth. To update $\Theta$ parameterized by $\theta$, we utilize SDS loss, which calculates per-pixel gradients by computing the difference between predicted noise and added noise as follows:

$$\nabla_\theta \mathcal{L}_{SDS}(\psi, \boldsymbol{x}) = \mathbb{E}\left[w(t)(\boldsymbol{\epsilon}_\psi(\boldsymbol{z}_t; \boldsymbol{y}, t) - \boldsymbol{\epsilon})\frac{\partial \boldsymbol{z}}{\partial \boldsymbol{x}}\frac{\partial \boldsymbol{x}}{\partial \theta}\right] \tag{3}$$

where $\psi$ parameterizes multi-view aware image diffusion model, $\boldsymbol{x}$ represents the RGB rendering output, $w(t)$ signifies a weight function for different noise levels, $\boldsymbol{z}_t$ denotes the latent encoding of $\boldsymbol{x}$ with the addition of noise $\boldsymbol{\epsilon}$, and $\boldsymbol{\epsilon}_\psi$ is the predicted noise with reference image $\boldsymbol{y}$ and noise level $t$.

**Auxiliary objective terms.** We leverage several additional loss terms to aid in the optimization process. To promote the consistency between the reference image and output 3D reconstruction, we introduce the color loss between the reference image $I_{ref}$ and the rendering from the reference viewpoint $\boldsymbol{x}_{ref}$ as follows:

$$\mathcal{L}_{ref} = \|I_{ref} - \boldsymbol{x}_{ref}\|_1. \tag{4}$$

Similar to the color loss, we also leverage the mask loss, which compares the mask of the reference image with the mask of the rendering to promote shape consistency, as follows:

$$\mathcal{L}_{mask} = \|M(I_{ref}) - M(\boldsymbol{x}_{ref})\|_1, \tag{5}$$

where $M$ is the masking function used for binary separation between the object and the background.

To impose regularization on the mesh surface, parameterized by SDF representations, we employ SDF regularization methods akin to those proposed by Liao et al. (2018) and (Li et al., 2023). Utilizing the binary cross entropy $BCE$, sigmoid function $\sigma$, and the sign function $\text{sign}$, we can express the first SDF regularizer as follows:

$$\mathcal{L}_{reg} = \sum_{i,j \in \mathbb{S}} BCE\Big(\sigma(s_i), \text{sign}(s_i) + BCE(\sigma(s_j), \text{sign}(s_i))\Big), \tag{6}$$

where $s_i$ is the SDF at the vertex $v_i$ and $\mathbb{S}$ is set of unique edges.

To further encourage the smoothness of reconstructed surface, we regularize the mean curvature of SDF which is computed from discrete Laplacian. The Laplacian loss is defined as:

To additionally promote the smoothness of the reconstructed surface, we regularize the mean curvature of SDF, computed from discrete Laplacian. The Laplacian loss is formally defined as:

$$\mathcal{L}_{lap} = \frac{1}{N} \sum_{i=1}^N |\nabla^2 s_i|. \tag{7}$$

The overall loss can be defined as the combination of $\mathcal{L}_{SDS}, \mathcal{L}_{ref}, \mathcal{L}_{mask}, \mathcal{L}_{reg}$ and $\mathcal{L}_{lap}$.

## 3.3 NEURAL SURFACE RECONSTRUCTION WITH NEURAL PBR TEXTURE

We employ DMTet in conjunction with a physically-based rendering (PBR) material model (McAuley et al., 2012), as depicted in (Munkberg et al., 2022). This choice allows us to incorporate spatially-varying Bidirectional Reflectance Distribution Function (BRDF) modeling for textures. The material model generates textures that vary spatially and comprises three key components: diffuse lobe parameters $k_d \in \mathbb{R}^3$, the roughness and metalness term $k_{rm} \in \mathbb{R}^2$, and the normal variation term $k_n \in \mathbb{R}^3$. The specular highlight color, denoted as $k_s$, can be determined with diffuse value $k_d$ and the metalness factor $m$ with the formula $k_s = (1 - m) \cdot 0.04 + m \cdot k_d$. It enables us to achieve photorealistic surface rendering and enhances the potential of diffusion models for improved realism.

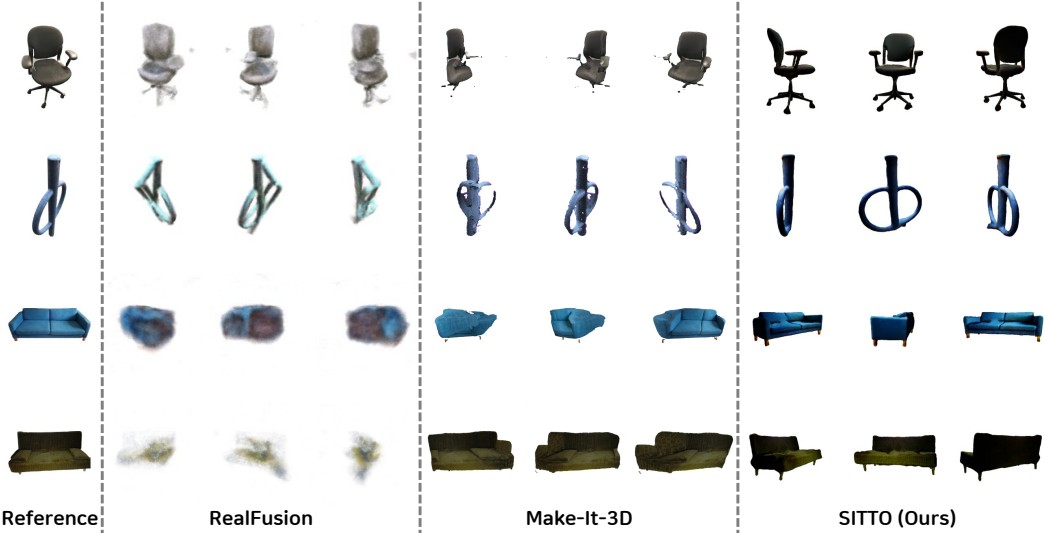

Figure 4: **Comparison of novel view rendering.** SITTO exhibits fine-grained geometry details and realistic reconstructions of the texture for novel views, compared to counterpart works.

| Method | Reference View | | | Novel Views | |
|---|---|---|---|---|---|
| | LPIPS ↓ | PSNR [dB] ↑ | CLIP Score ↑ | CLIP Score ↑ | min. CLIP Score ↑ |
| RealFusion | 0.1809 | 21.56 | 0.8494 | 0.7538 | 0.7030 |
| Make-it-3D | 0.0867 | 22.45 | 0.9386 | 0.8937 | 0.8046 |
| SITTO (ours) | **0.0777** | **22.89** | **0.9465** | **0.8942** | **0.8286** |

Table 1: **Comparisons of texture reconstruction and perceptual quality.** We measure the LIPIS, PSNR, and CLIP Score at the reference view and novel views. For the comparison of novel view renderings, we render the same number and location of viewpoints for all methods.

## 4 EXPERIMENTS

We show our high-fidelity textured mesh reconstruction results qualitatively and quantitatively in Sec. 4.1 and Sec. 4.2, respectively. We demonstrate the necessity of the initial shape prediction and viewpoint estimation within the *Image-to-Shape* pipeline in Sec. 4.3.

### 4.1 QUALITATIVE ANALYSIS

We present and assess the quality of reconstructed 3D textured meshes in terms of geometric and appearance attributes. We conduct a comparative analysis involving two relevant works: RealFusion (Melas-Kyriazi et al., 2023) and Make-it-3D (Tang et al., 2023), where . they are the most recent advancements in this field. In Fig. 4, we showcase SITTO's fine-grained geometry details and realistic texture qualities of the reconstructed meshes for novel view renderings. While competing methods severely fail to reconstruct realistic and interpretable 3D geometries or textures, our SITTO reconstructs 3D meshes with realistic geometry and consistent textures even for unseen viewpoints.

### 4.2 QUANTITATIVE ANALYSIS

We also conduct comparisons using quantitative metrics to assess the quality of textured mesh reconstruction and the rationality of geometric properties.

**Textured 3D mesh reconstruction.** We first evaluate the reconstruction quality of textures with the counterparts: RealFusion (Melas-Kyriazi et al., 2023) and Make-it-3D (Tang et al., 2023). As depicted in Table 1, we measure the similarity between the reference image and the rendered image

| Metric | Total3D | Total3D (w/o refine) | Mesh R-CNN | SITTO (ours) |
|---|---|---|---|---|
| Chamfer Distance ↓ | 0.0322 | 0.0353 | 0.1261 | 0.0872 |

Table 2: **Comparisons of mesh reconstruction.** We measure the Chamfer distance between point clouds uniformly sampled from the predicted and ground-truth mesh. Point clouds are normalized in scale and aligned to the ground-truth point clouds by the iterative closest point (ICP) algorithm. 10K points are sampled for each mesh.

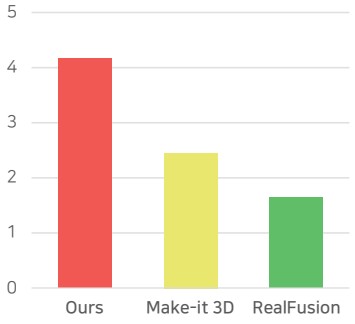

(a) Quality of the reconstructed textured mesh.

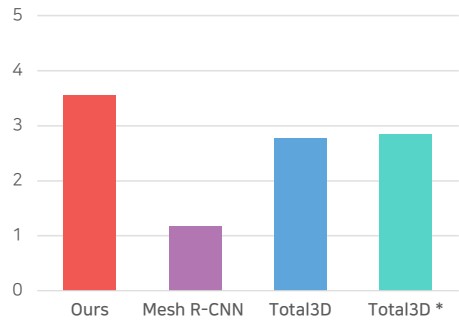

(b) Rationality of the reconstructed mesh geometry.

Figure 5: **Human perceptual studies.** We asked 30 users to evaluate (a) the quality of the reconstructed textured mesh and (b) the rationality of the reconstructed mesh geometry. SITTO(ours) outperforms other competing methods in both aspects. The notation Total3D[*] denotes that we use the coarse shape predicted from the Total3D, by skipping the mesh refinement stage.

at the reference view and novel views, respectively. We use three metrics to assess these aspects: 1) peak signal-to-noise ratio (PSNR), 2) learned perceptual image patch similarity (LPIPS) (Zhang et al., 2018), and 3) contrastive language-image pre-training (CLIP) score (Radford et al., 2021), which evaluates the semantic similarity in the vision-language aligned space. To see the consistency of the appearance between novel views, we also report the minimum value of the CLIP score. Our method outperforms the prior arts in terms of both reference-view reconstruction and novel-view rendering qualities. It demonstrates proficiency in reconstructing 3D geometry and appearance and preserving the semantic content of our method. Our SITTO takes less than an hour as we directly optimize the 3D textured mesh; otherwise, counterparts take more than two hours with NeRF representation. SITTO exhibits a notable efficiency advantage, requiring less than an hour to complete textured mesh optimization. Our time consumption contrasts comparable methods, which employ the NeRF representation and require more than two hours but are still limited in visual quality.

**Rationality of geometry.** To focus on the reconstructed 3D geometry quality of our SITTO, we further assess geometry reconstruction quality with the shape (*i.e.*, textureless mesh) prediction methods (Gkioxari et al., 2019; Nie et al., 2020). For this, we evaluate the Chamfer distance of sampled points between the ground-truth mesh and output mesh of each method. Note that our optimization process does not access the ground-truth 3D information, *e.g.*, point clouds, voxels, and meshes, while both Mesh R-CNN and Total3DUnderstanding are directly trained with Chamfer distance from ground-truth meshes as supervision. Despite this, as shown in Table 2, SITTO outperforms geometry reconstruction than Mesh R-CNN (Gkioxari et al., 2019). Total3DUnderstanding surpasses ours; however, the reconstructed meshes tends to be incomplete and show degenerated geometry compared to ours, especially in the perceptual aspect. It means that the practical utility of generated meshes is limited in real applications. In contrast to prior approaches that mimic rigorous 3D ground-truth mesh templates, we emphasize achieving a holistic and perceptually reasonable reconstruction of 3D textured meshes. It is worth noticing that SITTO also reconstruct image-aligned and realistic textures, where Mesh R-CNN and Total3DUnderstanding are limited in reconstructing only 3D shapes.

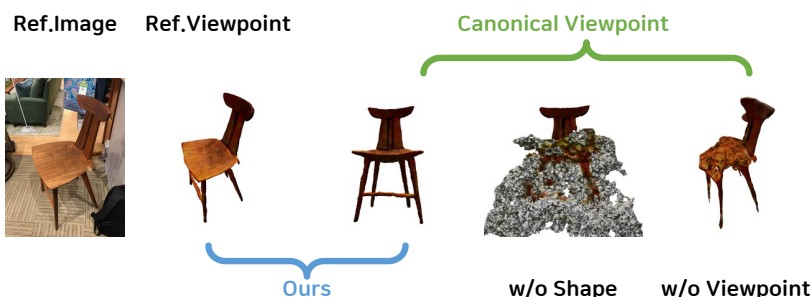

Figure 6: **Ablation studies.** To validate the efficacy of our pipeline design, we perform ablation studies in scenarios where the initial shape or viewpoint estimation is absent. Canonical viewpoint means that the camera location demonstrated in Fig. 3a.

### 4.3 Additional Studies

**Human perceptual studies.** In this section, we assess previously unexamined aspects by employing quantitative measurements in Sec. 4.2 and two human evaluations about textured mesh and geometry reconstruction quality from a perceptual perspective. Firstly, we evaluate the overall consistency and perceptual quality, encompassing aspects of both geometry and appearance. In Fig. 5a, our approach consistently outperforms other methods, consistent with the quantitative results presented in Sec. 4.2. Secondly, we specifically focus on assessing the quality of geometry reconstruction. While our approach may not achieve the highest scores in geometric metrics like Chamfer distance, human perceptual evaluations indicate that our reconstruction exhibits superior perceptual quality and geometric completeness compared to counterpart methods, as illustrated in Fig. 5b.

**Ablation studies.** We conduct ablation studies of SITTO to better understand how design choices impact our approach. In the first scenario, where the initial shape is omitted, the optimization process encounters challenges in achieving accurate surface representations, particularly when it comes to flat surfaces. In the second scenario, where viewpoint estimation is omitted, the camera is positioned as shown in Fig. 3a. Noticeably, the viewpoint estimation module plays a pivotal role in optimizing fine-grained geometry details. Without reference viewpoints, the misalignment between the reference image and default viewpoints often leads to inconsistent and degenerated geometry.

## 5 Conclusion

In this work, we present SITTO, a monocular 3D textured mesh reconstruction with generative test-time optimization. Our approach addresses several challenges in reconstructing a 3D textured mesh from a single image. First, we highlight the limitations of feed-forward-based shape prediction methods, which often struggle to ensure high-quality mesh estimation results. Second, we emphasize the necessity of estimating the viewpoint of the given single image to leverage the capacity of multi-view image diffusion models fully. To overcome these challenges, we propose the method that combines the predictions of a model capable of estimating both mesh and viewpoint with a test-time optimization approach using multi-view diffusion models. This enables us to achieve fine-grained geometry and photorealistic texture appearance. Importantly, our method conducts optimization solely through mesh representation, efficiently using both time and memory resources while obtaining high-resolution 3D textured meshes. We believe that our approach represents a significant step towards democratizing single-image-to-3D-content generation and improving the overall quality of results in this fields.

### Reproducibility Statement

We will make our code accessible to the public once it is published. Additionally, for convenience, we will provide a demo code along with configuration details.

ETHICS STATEMENT

The task of 3D textured mesh reconstruction involves comprehending the geometry and visual characteristics of objects. Our model excels in reconstructing highly detailed 3D shapes and appearances from single images in real-world scenarios, offering practical applications in 3D interior design, content generation, and augmented reality.

However, it is important to acknowledge potential ethical concerns, such as the possibility of generating misleading media or enabling design theft through the duplication of existing physical products.

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

# A APPENDIX

## A.1 EXPERIMENTAL SETUP

We provide details regarding our experimental setup, including the dataset and implementation details.

**Datasets.** To evaluate the 3D reconstruction quality from the image, we make a test subset which consists of several image-3D mesh pairs sampled from Pix3D (Sun et al., 2018). Because the images of Pix3D are not intensively curated in that they include scenes occluded by text or humans, and even screenshots from online shopping platforms, we carefully reject those samples in our test subset. Despite our *Image-to-Shape* architecture choice, *i.e.*, Total3DUnderstanding, which is trained for Pix3D only, our pipeline is working for real-world scene images to some extent. We also acquire RGB images, captured by an iPhone, to test the real-world scenario.

**Implementation details.** We optimize geometry and texture concurrently thanks to initial shape and powerful multi-view diffusion guidance. We use AdamW optimizer with gradient clipping and the respective learning rates of $1 \times 2$ for geometry and $1 \times 3$ for texture. For each iteration, we randomly sample 8 camera viewpoints for rendering the novel views. We conduct training with one NVIDIA A6000 GPU in less than one hour. We leverage open3d (Zhou et al., 2018) to deal with SDF and point cloud representations.

## A.2 ADDITIONAL QUALITATIVE RESULTS

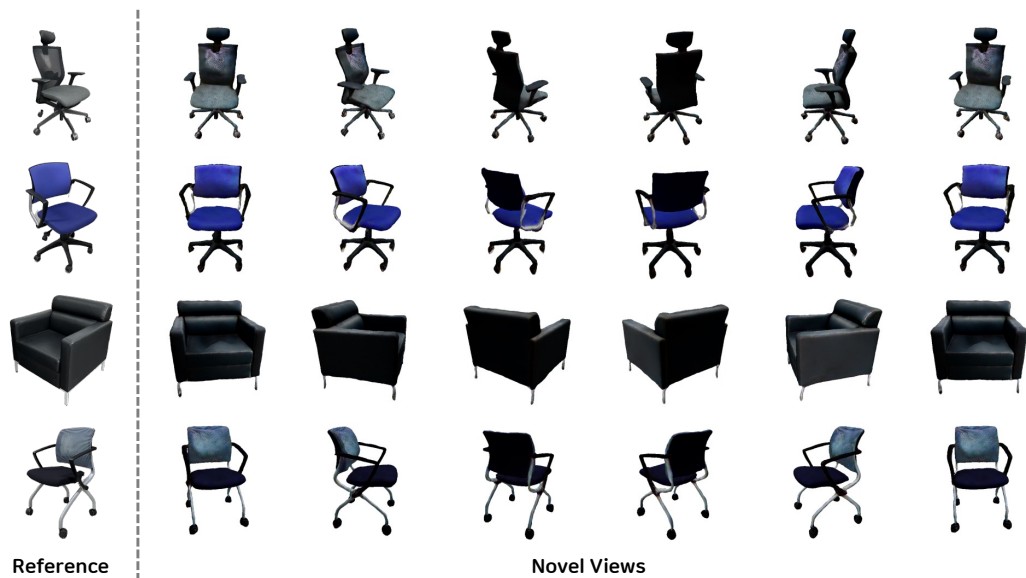

Reference          Novel Views

Figure 7: **Qualitative results of real data.** We acquire real data from a real scene, which is not Pix3D datasets. Reference means the given image image, which is used for the condition of multi-view diffusion. We can generate high-fidelity geometry and realistic texture rendering.

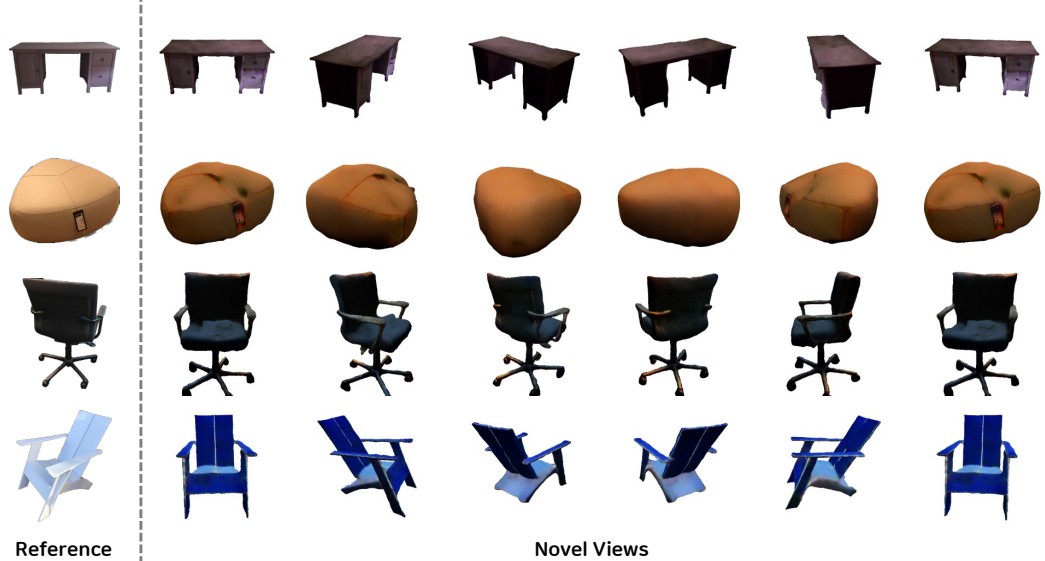

Figure 8: **Qualitative results of Pix3D data.**

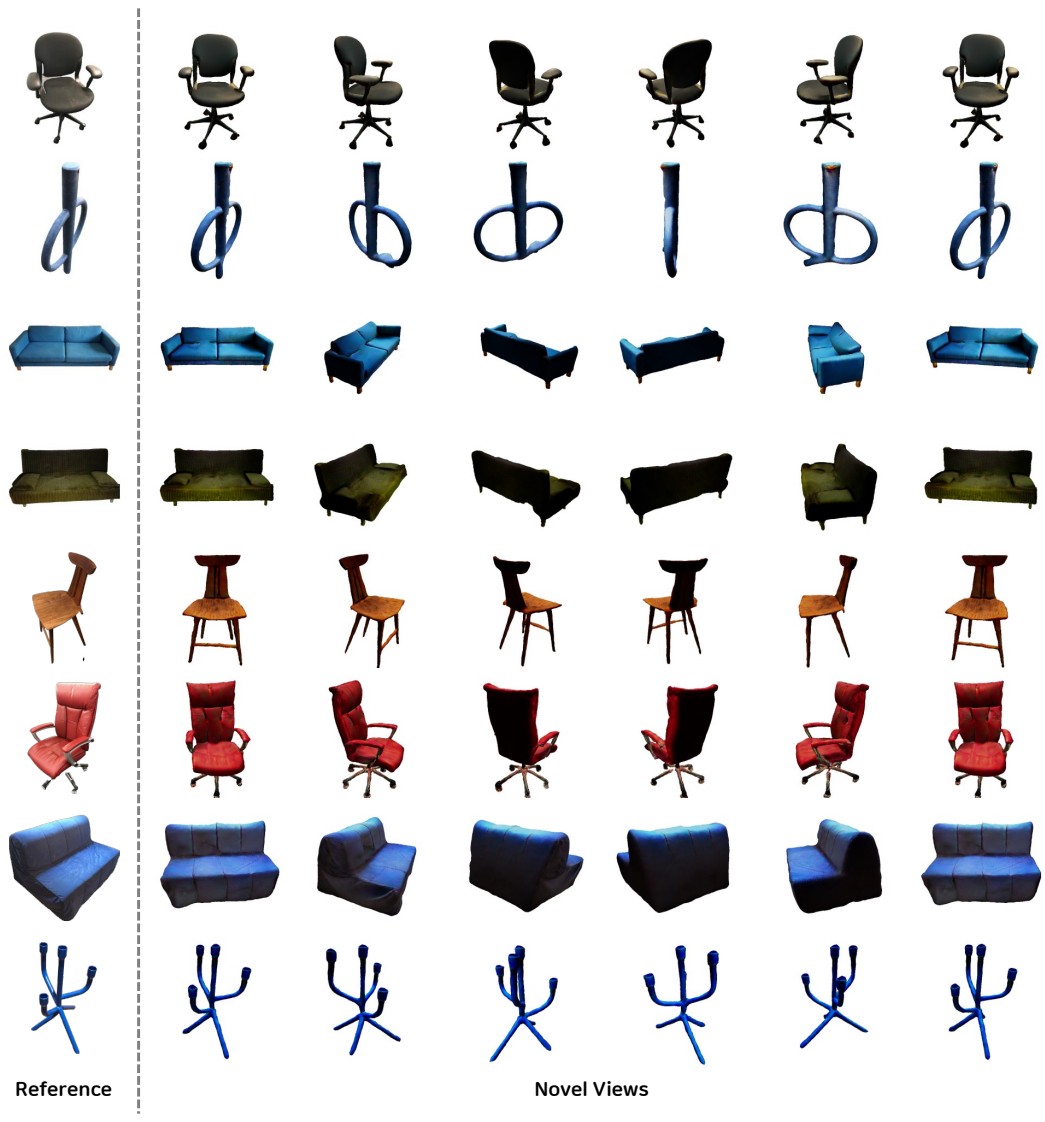

Reference                                          Novel Views

Figure 9: **Qualitative results of Pix3D data.**

