# OpenReview forum: "SITTO: Single-Image Textured Mesh Reconstruction through Test-Time Optimization"
_ICLR.cc/2024/Conference — ICLR 2024 Conference Withdrawn Submission_

### Official Review · Reviewer_dbEu · 2023-10-21

**Soundness:** 1 poor
**Presentation:** 2 fair
**Contribution:** 1 poor
**Rating:** 3
**Confidence:** 5

**Summary:**

This paper tackles the problem of monocular 3D textured mesh reconstruction with test-time optimization. The authors propose a framework that integrates feed-forward mesh generation that can quickly create 3D initial shapes and an optimization-based feedback loop in test time.

**Strengths:**

- The integration of feed-forward mesh generation and test-time optimization is reasonable.
- This paper is easy to follow.

**Weaknesses:**

- The novelty of this paper is limited. It just mounts the optimization block based on Zero123 SDS loss onto a feed-forward mesh generation method, which is a well-used setup.
- The experiments are not performed well. The experiments are only performed on chairs, and the compared methods are few. Both the qualitative experiments and quantitative experiments are not enough to demonstrate the effectiveness of the proposed method.

**Questions:**

The main body of this paper seems to exceed 9 pages?

---

### Official Review · Reviewer_JToQ · 2023-10-28

**Soundness:** 2 fair
**Presentation:** 2 fair
**Contribution:** 2 fair
**Rating:** 3
**Confidence:** 5

**Summary:**

This paper introduces a technique for generating 3D shapes from a singular image. Initially, it utilizes Total3DUnderstanding to deduce the initial shape and viewpoint from a single image and incorporates a pre-existing segmentation module for image segmentation. In the subsequent phase, the initial mesh is refined using DMTet and the SDS loss, derived from a pretrained 2D diffusion network, Zero123. Multiple loss functions are used to aid the optimization process.

**Strengths:**

This paper introduces an innovative technique for deriving 3D shapes from a single image. Employing a feed-forward approach to produce an initial shape may simplify the optimization process and enhance efficiency.

**Weaknesses:**

1. The motivation behind the proposed two-stage pipeline remains unclear. While the per-shape optimization via Zero123 can accommodate various categories in an open-world scenario, the integration of a feed-forward method for the initial shape generation might compromise this capability, since the feed-forward module is trained on a finite 3D dataset.

2. In the evaluation results, the focus appears to be on familiar categories like chairs, sofas, and tables. If the methodology primarily targets closed domain 3D generation, there are more robust baselines available, such as LAS-Diffusion and Get3D, which excel in these common categories. However, comparisons with these are conspicuously absent. The method seems to benchmark only against a few optimization-based techniques, which are known to proficiently handle open-world objects.

3. The experimental evaluation is very weak. The training and test set are not clearly stated. It's unclear how many shapes are used in the quantitative evaluation and whether the construction of the test set is fair.

3. The experimental evaluation lacks depth. The specifications of the training and test sets are vague. There's ambiguity regarding the number of shapes used for quantitative analysis and whether the test set is constructed fairly.

4. The user study section lacks clarity. It would be beneficial to elaborate on the details, such as the number of shapes evaluated and the methodology behind score computation.

5. The ablation study is weak and problematic.  Without the initial shape, the per-shape optimization with Zero123 should yield superior outcomes. Additionally, when using Zero123 with the SDS loss, there's no apparent need for camera poses.

6. The document would benefit from meticulous proofreading. Notable typos include:

a. page 1: "which, as explained in ?, imposes constraints on the resulting resolution."

b. page 7: "where . they are the most recent advancements in this field."

**Questions:**

1. In equation (2), could you clarify the method used to sample the point set P?

2. Equation (6) is not straightforward, and there appears to be a mismatch in the parenthesis. An elaboration on the rationale behind this loss function would be greatly appreciated.

3. The equation: "ks = (1 − m) · 0.04 + m · kd." requires a more detailed explanation.

4. On page 8, the statement, "To assess the appearance consistency between novel views, we also present the minimum value of the CLIP score," is ambiguous to me.

---

### Official Review · Reviewer_tSLs · 2023-11-01

**Soundness:** 1 poor
**Presentation:** 1 poor
**Contribution:** 1 poor
**Rating:** 1
**Confidence:** 5

**Summary:**

The paper proposes SITTO, a two-stage method for single-image 3D reconstruction. For the first stage, it uses a traditional domain-specific single-image mesh reconstruction model Total3DUnderstanding, which is trained on Pix3D (395 furniture models). For the second stage, it leverages Magic3D-style SDS optimization for mesh refinement.

**Strengths:**

The model could generate some high-quality furniture shapes.

**Weaknesses:**

1. Spending 1 hour to optimize a coarse mesh from a domain-specific model for furniture is not necessary. For domain-specific single-image 3D reconstruction, there are many existing fast and robust models—for example, Single-Stage Diffusion NeRF: A Unified Approach to 3D Generation and Reconstruction.
2. No technical novelty. It is a simple combination of two off-the-shelf models: Total3DUnderstanding and Magic3D-style DMTet finetuning.
3. The domain-specific model is trained on Pix3D. And the experiments are conducted on Pix3D. Such comparisons to those zero-shot single-image 3D reconstruction models are even more unfair.
4. No two-stage ablation studies. The paper does not ablate either stage to show the significance of design choices.

**Questions:**

Please try conducting ablations on two stages and evaluating the models on a more diverse set of shapes.

There is no discussion of limitations. The current image-to-shape model cannot be generalized outside a limited amount of furniture.

I recommend that authors refrain from haphazardly combining A and B without a well-founded motivation for the task. It is crucial to conduct more comprehensive literature reviews to provide a solid foundation for the task and ensure a strong motivation for your research.

Writing mistakes:

1. Parenthetical citations are not used correctly in many cases.
2. Page 2, “explained in ?” is not specified.
3. Sec 4.1, “where . they are”